behaviour

recruitment, collective behaviour, nest entrance, ants, foraging, trail

**Author for correspondence:**
Bertrand Collignon
e-mail: bertrand.collignon@ulb.ac.be

# Multiple nest entrances alter foraging and information transfer in ants

Marine Lehue[1], Bertrand Collignon[1,2] and Claire Detrain[1]

[1]Université Libre de Bruxelles, Bruxelles, Belgium
[2]Ecole Polytechnique Fédérale de Lausanne, Lausanne, Switzerland

ML, 0000-0002-9276-909X; BC, 0000-0003-2463-1512

The ecological success of ants relies on their ability to discover and collectively exploit available resources. In this process, the nest entrances are key locations at which foragers transfer food and information about the surrounding environment. We assume that the number of nest entrances regulates social exchanges between foragers and inner-nest workers, and hence influences the foraging efficiency of the whole colony. Here, we compared the foraging responses of *Myrmica rubra* colonies settled in either one-entrance or two-entrance nests. The total outflows of workers exploiting a sucrose food source were similar regardless of the number of nest entrances. However, in the two-entrance nests, the launching of recruitment was delayed, a pheromone trail was less likely to emerge between the nest and the food source, and recruits were less likely to reach the food target. As a result, an additional entrance through which information could transit decreased the efficiency of social foraging and ultimately led to a lower amount of retrieved food. Our study confirms the key-role of nest entrances in the transfer of information from foragers to potential recruits. The influence of the number of entrances on the emergence of a collective trail also highlights the spatially extended impact of the nest architecture that can shape foraging patterns outside the nest.

## 1. Introduction

In animal societies, collective decision-making and coordinated behaviours often arise from multiple interactions between group-mates, even though each individual has access only to local and partial information and is not aware of the global pattern that is emerging [1–5]. In these processes, the coupling of positive and negative feedbacks that influence local rates of interactions between nest-mates, contributes to generate an amazing diversity and complexity of adaptive structures at the colony level.

Regarding positive feedbacks, any signal that is emitted by one individual may elicit a behavioural response in receivers

which may, in turn, produce a signal identical to the one received [6]. Above a certain threshold of interaction rates, a local behaviour and/or information is amplified and is ultimately propagated to the whole group [7,8]. In the well-known case of collective foraging by ants, one of these positive feedbacks consists of a recruitment pheromone laid on the ground by individuals that have found a profitable food source [9–11]. The social amplification of this information eventually leads to a collective exploitation of the food target, using a well-defined network of foraging trails [12,13].

Another important positive feedback known to regulate collective foraging activity in ants is the interaction rate between successful foragers and potential recruits that are present inside the nest [14–17]. In this case, the ant nest plays a key role in shaping interactions between these informed and naive colony members. Indeed, the nest structure is known to shape the local density of hosted nest-mates and thereby to regulate the social transmission of information about discovered resources [18–21]. In particular, the chamber connected to the nest entrance, 'the entrance chamber', often hosts large aggregates of workers. While the entrance chambers are key locations where workers can be recruited for the defence of the colony [22,23], can reject intruders [24] or can check sanitary risks associated with diseased nest-mates or contaminated items [25], they are also a crucial place for food exchange and information sharing [16,17,19,26–28]. In addition, because most of the ants located close to the nest openings are oriented towards the incoming foragers, the entrance area acts as a front line where information about environmental opportunities can be first processed and easily updated by nest-mates [29]. At this particular hotspot for information transfer, it has been demonstrated that the rate at which recruits leave the nest depends on the rate at which successful foragers carry food and provide antennal contacts [15–17,26], but also on the design of the entrance [30]. In this respect, the number of nest entrances may shape the food recruitment process by segregating the information flow, thus altering the encounter rates that are locally experienced by potential recruits inside the nest. One may expect that the structure of this physical interface between the nest and the environment will deeply influence the foraging strategies displayed by the ant colony as a whole.

So far, most of the studies about ant foraging have been carried out in laboratory-housed colonies operating from a single nest entrance. However, in the field, it is common to find ant species that inhabit nests showing multiple entrances. This is the case for natural nests of the red ant *Myrmica rubra*, that can show from one to six nest entrances being separated by a few centimetres up to a few decimetres (M. Lehue 2018, personal field observations). In this paper, we investigated to what extent an additional nest entrance could influence the collective food recruitment and the patterns of foraging trails displayed by *M. rubra* ant colonies. Preliminary baiting experiments showed that foragers exiting from different entrances could exploit the same food source in the field. Therefore, we compared the recruitment dynamics towards a single food source for ant colonies that were housed either in one- or two-entrance nests. For both conditions, we studied the ability of recruiters to mobilize nest-mates out of the nest and the patterns of foraging trails connecting the nest to the food target. At the collective level, we quantified the mobilization of the foragers and followed their spatial distribution in the foraging arena using an automated image analysis to detect the emergence of foraging trails. Then, at the individual level, we quantified the proportion of recruits that successfully reached the food source, by tracking focal workers in the outgoing flow of foragers. Finally, we investigated the impact of the number of nest entrances on the global efficiency of the ant colony to exploit a sugar food solution.

# 2. Material and methods

## 2.1. Ant colonies

*Myrmica rubra* is a polydomous, polygynous and monomorphic ant species that is common in European temperate areas. This ant species lives in semi-humid conditions and can be found in biotopes such as semi-open grasslands or paths' borders [31]. *Myrmica rubra* nests show multiple entrances and are built in various substrates allowing a good humidity, under stones, inside rotting wood, among roots of nettles and bramble bushes [32]. Nine colonies of *M. rubra* were excavated from earth banks in a semi-open grassland located in Aiseaux and Falisolle (Belgium) in June 2016. Once in the laboratory, we reared ant colonies in artificial nests placed in foraging arenas, the walls of which were coated with Fluon (Whitford, UK) to prevent ant escape. We kept laboratory conditions at $21 \pm 0.4°C$ and $52 \pm 2\%$ relative humidity, with a constant photoperiod of 12 h a day.

| experiment 1 | experiment 2 |
|---|---|

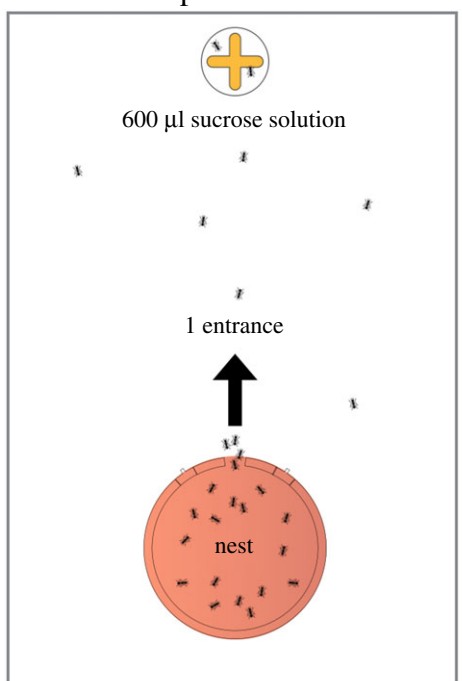
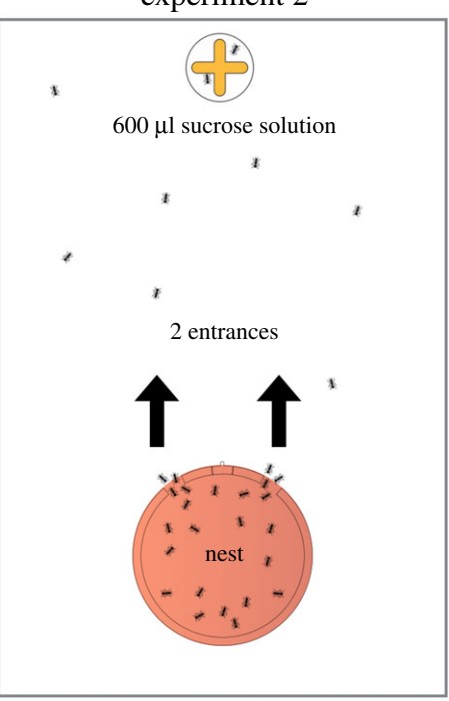

**Figure 1.** Diagram of the experimental set-up. Colonies were housed in nests of which one entrance or two entrances were open. Nests were placed on one side of the arena and a plate containing the sucrose solution on the opposite side.

## 2.2. Experimental set-up

Experimental nests were made out of a laser-cut Plexiglas circular wall covered with a Plexiglas ceiling. Internal dimensions of the circular nests were 8 cm diameter and 2 mm high. Each of the nine experimental colonies contained one queen, 300 workers and brood that covered approximately 10% of the nest area. The nest included three entrances (each 10 mm wide and 5 mm long) placed 15 mm apart from each other (figure 1). We used two different nest entrance configurations in the experiments: either one entrance—the central entrance—was open, or two entrances—the two lateral entrances—were open. We closed entrances using laser-cut fitting pieces of cardboard. We placed the experimental nest on one side of a rectangular arena (45 × 30 cm) as shown in figure 1. We covered the floor of the arena with plaster and daily watered around the nest to provide the humidity necessary to the ant survival. Before the start of the experiments, we moved colonies into these experimental nests, where they could acclimatize for 48 h.

## 2.3. Experimental procedure

In order to assess how the number of nest entrances may influence recruitment and food harvesting, we tested each colony in nests of which either one entrance or two entrances were open (each colony underwent the two treatments successively). Each nest configuration was tested in a randomized order. Before each experiment, ants were starved for 48 h and could only access a water tube placed on the opposite side of the arena. On the experimental day, we placed in the arena a feeder consisting of 600 µl of 1 M sucrose solution. The feeder plate was cross-shaped in order to increase its perimeter/area ratio, thereby preventing congestion effects around the food source during recruitment. We positioned the feeder, centred and equidistant to the two lateral nest entrances, at a distance of 27 cm from the central nest entrance. The experiment lasted 120 min during which we filmed the entire arena using Logitech C920 webcams (1920 × 1080 pixels resolution, 15 fps). At the end of the experiment, we removed the food plate and changed the configuration of the nest entrances. Colonies rested for 4 days in the new nest configuration and had access ad libitum to water, a 0.3 M sucrose solution, and dead *Tenebrio molitor* mealworms. Then, the colonies underwent the second experiment in this new nest configuration.

## 2.4. Worker mobilization

On video recordings, we quantified the level of workers' mobilization in each nest configuration. First, we calculated (i) the number of scouting ants present in the arena at the beginning of each experiment, prior to the introduction of the food source, (ii) the baseline outflow of workers for 10 min before the introduction of the food source, and (iii) the density of ants present in the entrance area (i.e. within a 2 cm area from the entrance as shown in [29]). Once we introduced the food source, we measured (iv) the outflow of ants per 5 min over the whole course of the experiment, which gave us the total number of foraging trips during the 120 min foraging period. Furthermore, we measured the outgoing flows every minute during the first 30 min in order to get a closer follow-up of ant mobilization when the recruitment was the strongest. Out of these data, we computed (v) the maximal outflow per minute and (vi) the time at which this maximal outflow occurred. As our results were paired, measures (i)–(vi) were compared between the two nest configurations using Wilcoxon signed-rank tests.

For the two-entrance nest configuration, we also computed an index of asymmetry ($I_a$) to see whether the outflows of ants were evenly distributed between the two open doors. This index was computed as follows:

$$I_a = \left| \frac{T_{FL} - T_{FR}}{T_{FL} + T_{FR}} \right|,$$

with $T_{FL}$ and $T_{FR}$ being the total outflow of ants through the left and right entrance, respectively. This index varied between 0 for a perfectly symmetrical use of both entrances and 1 when a single entrance was used by outgoing ants. This index was computed for the early stage of foraging ($I_{a5\,min}$, from $t = 0$ to $t = 5$ min) and over the course of the whole experiment ($I_{a120\,min}$, from $t = 0$ to $t = 120$ min). In addition, we assessed whether the mobilization of recruits was taking place independently at each entrance or if the two outgoing flows of foragers were synchronized. To do so, we computed an index of synchronization $I_s$ as follows: every minute, the differences between the outgoing flows observed at the two entrances were calculated and their absolute values were summed for the first 30 min of the experiment. This is described in the following equation:

$$I_s = \sum_{t=1}^{30} |F_{Lt} - F_{Rt}|,$$

with $F_{Lt}$ and $F_{Rt}$ being the 1 min outgoing flow at time $t$ at the left and right entrance, respectively. Two perfectly synchronized flows would result in a value of $I_s = 0$. If the flows at the two entrances tend to diverge, this would result in an increase of the index $I_s > 0$. We performed permutation tests to determine whether the observed levels of synchronization were statistically significant or did not differ from the ones expected by chance. To do so, we compared the observed experimental synchronization index with 10 000 simulated indexes. The 1 min flows $F_{Lt}$ and $F_{Rt}$ measured at the two entrances during the first 30 min of experiments were randomly permuted 10 000 times and an index was computed for each permutation. Then, for each replicate of the experiment, we computed the $p$-value of the synchronization index as the proportion of simulated indices that were smaller than the experimental index value. Because one permutation test was performed for each replicate of the experiment, we used Bonferroni correction to control for multiple comparisons.

## 2.5. Emergence of a foraging trail

We investigated whether the number of nest entrances could alter the emergence of a well-defined chemical trail leading to the food source. Instead of relying on a by-eye trail detection that could be individually biased, we designed a technique of image processing that allowed us to visualize the dynamical formation of trail routes. Each 120 min video-recording was divided into 24 periods of 5 min each. Then, each of these periods was condensed into a single image where the detected movements of all foragers were overlaid. In those images, the pixel intensity was directly related to the level of ant traffic, going from dark (low traffic intensity) to light pixels (high traffic intensity). The presence of a collective foraging trail was assessed by a two-step process. First, a decision tree based on the general characteristics of the images classified these images into 'candidate images' with a potentially structured foraging trail or 'rejected images' without structured foraging trail. Several image features were considered: the total intensity of the image, the spatial distribution of the pixel intensity and the variability in pixel intensity. Second, a ridge detector identified, on the candidate images, elongated structures of high intensity of pixels corresponding to intense ant traffic along a foraging trail and

classified them into structured foraging or rejected them as non-structured foraging. More details of the algorithm are presented in the electronic supplementary material, S2. Using this automated classification of trails, we obtained the time at which trails first emerged as well as their duration over the course of the experiment. For each colony, those values were compared between the two nest configurations using a Wilcoxon signed-rank test. We assessed whether colonial identity influenced the motivation to lay a trail using Spearman's correlation between trail duration in one and two-entrance nests.

## 2.6. Ability to reach the food source

For the two nest configurations, we performed an individual tracking of recruited individuals to assess their likelihood of reaching the food source. The tracking of recruits was done on video recordings and started 30 min after food introduction, once the recruitment was well-established. In total, we tracked 90 ants (10 ants per colony) for each nest configuration. To avoid bias owing to local group effects among ants that simultaneously exited the nest, we tracked one ant every five outgoing ants. In the two-entrance nest configuration, we tracked five ants exiting the left entrance and five ants exiting the right entrance. Each ant was followed for a maximum duration of 3 min. At the end of the 3 min observation, ants could have reached the food source, gone back to the nest or kept strolling in the nest surroundings. We used generalized linear mixed models (GLMM) assuming a binomial error distribution and logistic link function to analyse the impact of a second entrance on the destination of the foragers. To do so, a destination was tested against the two others with a dichotomous status (e.g. 1: reaching the food source; 0: back to the nest or strolling in the environment). This procedure was repeated for each of the three possible destinations. We included the number of entrances as fixed factor and colony identity as random factor.

## 2.7. Resource exploitation

In order to quantify the global efficiency of food exploitation, we measured the total consumption of sucrose solution at the end of 120 min of foraging. Food plates were weighted using a microbalance ($10^{-5}$ g accuracy; Metler Toledo AB125-S) at three different times: when the plate was empty, just after adding the 600 µl of sucrose solution at the start of the experiment, and after food consumption by the ants at the end of the experiment. We estimated the evaporation rate of the sucrose solution by placing two control food sources (same volume and sugar concentration) next to the experimental arena, and by weighing them at the end of the experiment. The observed evaporation rate was used to calculate the quantity of sucrose solution that was actually ingested by the ants. As the experiments were paired per colony, the sucrose consumption was compared between the two nests configurations using a Wilcoxon signed-rank test.

## 2.8. Fieldwork permissions

No permissions were required prior to conducting this research.

# 3. Results

## 3.1. Worker mobilization

Before the introduction of the food source, there was no difference between the 10 min total outflows of ants exiting either from a one-entrance or a two-entrance nest, which were of 33 [8; 54] and 33 [7; 69] ants, respectively (median [range], $n = 9$, $p = 0.84$, $W = 16$ Wilcoxon signed-rank test). Likewise, the ant densities in the entrance areas did not significantly differ between one and two-entrance nests and were respectively of 2.67 [2.14; 4.10] and 2.23 [1.87; 4.01] ants cm$^{-2}$ (median [range], $p = 0.41$, $n = 9$, $W = 30$ Wilcoxon signed-rank test). Once the food was discovered by scouts, the total outgoing flows steeply increased within the first 10 min of foraging and then tended to stabilize between 25 and 35 ants 5 min$^{-1}$ (figure 2$a$). After 2 h of food exploitation, the total number of foraging trips did not significantly differ between the two nest configurations ($n = 9$, $p = 0.95$ Wilcoxon signed-rank test, figure 2$b$). Over the whole course of the experiment, we observed a median [range] number of total foraging trips of 561 [394; 909] and 565 [464; 770] workers for one-entrance and two-entrance nests, respectively.

A detailed analysis of the outflows per minute revealed peak values that occurred during the first 30 min of foraging. Indeed, the outflows of ants momentarily reached similar maximal values for the two nest configurations, which were 15 [9; 37] and 16 [10; 43] ants min$^{-1}$ for the one-entrance and the two-

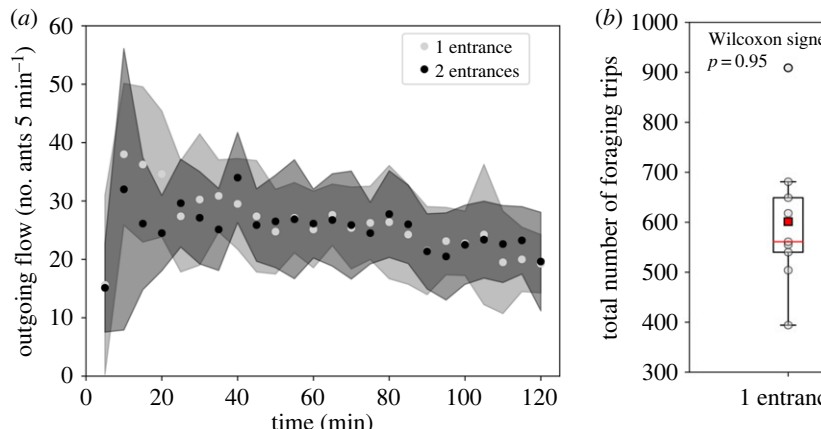
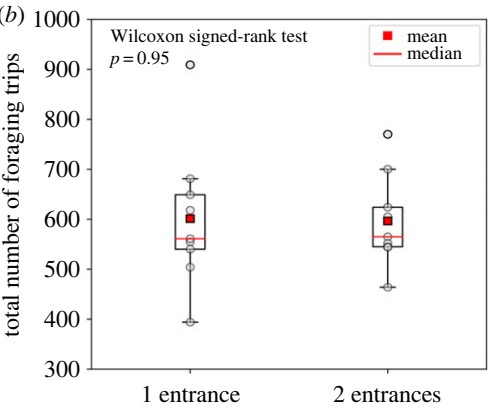

**Figure 2.** Influence of the number of nest entrances on the total outflow of ants. (*a*) Outflow dynamics of workers exiting the nest per 5 min of observation, in one-entrance nests (grey dots, light grey shading) and two-entrance nests (black dots, dark grey shading), Circles and shadings represent the mean ± s.d., $n = 9$. (*b*) Boxplots represent the distributions of the total outflow of ants per colony. The box extends from the lower to upper quartile values of the data, with a red line at the median. The whiskers extend from the box to show the range of the data ($n = 9$, Wilcoxon signed-rank test, $p = 0.95$).

**Table 1.** Comparison of foraging characteristics between one and two-entrance nests. (The asterisk indicates a statistically significant *p*-value.)

| | | one-entrance nests | two-entrance nests | test | *p* | test statistics |
|---|---|---|---|---|---|---|
| mobilization of workers | total outflow (median + range, *n* ants) | 561 [394; 909] ($n = 9$) | 565 [464; 770] ($n = 9$) | Wilcoxon signed-rank test | 0.95 | $W = 22$ |
| | max outflow/min (median + range, *n* ants) | 15 [9; 37] ($n = 9$) | 16 [10; 43] ($n = 9$) | Wilcoxon signed-rank test | 0.71 | $W = 19$ |
| | max outflow occurrence time (median + range, min) | 8 [3; 14] ($n = 9$) | 11 [6; 29] ($n = 9$) | Wilcoxon signed-rank test | 0.023* | $W = 1.5$ |
| pheromone trails | proportion of experiments with trails | 0.89 ($n = 9$) | 0.67 ($n = 9$) | Fisher exact test | 0.57 | odd ratio = 3.7 |
| | appearance time (median + range, min) | 12.5 [5; 50] ($n = 8$) | 25 [5; 35] ($n = 6$) | Mann–Whitney *U*-test | 0.21 | $U = 17.5$ |
| | trail duration (median + range, min) | 32.5 [5; 70] ($n = 8$) | 12.5 [5; 90] ($n = 6$) | Mann–Whitney *U*-test | 0.24 | $U = 18$ |
| resource exploitation | sucrose solution (median + range, mg) | 119 [94; 172] ($n = 9$) | 108 [69; 17] ($n = 9$) | Wilcoxon signed-rank test | 0.074 | $W = 38$ |

entrance nests, respectively (median [range], Wilcoxon signed-rank test, $p = 0.71$, $n = 9$; table 1). In the case of one-entrance nests, the maximal outflows of ants were observed at 8 [3; 14] min after food introduction. The mobilization of ants was significantly delayed in two-entrance nests, with a maximal outflow that

**Figure 3.** Example of trail detection. (*a*) Image processing of 5 min segments allowing the detection or not of a foraging trail. (*b*) Emergence of a foraging trail in one-entrance nest. (*c*) Individual trails were too dispersed to form a coherent collective foraging trail. Green rectangles indicate 5 min segments where a foraging trail was detected, red rectangles indicate segments where no foraging trails were detected.

took place after 11 [6; 29] min (median [range], Wilcoxon signed-rank test, $p = 0.023$, $n = 9$; table 1). While multiple entrances did not affect the total outflow of workers, they appeared to delay the recruitment dynamics and its amplification process.

In two-entrance nests, we also examined whether the flows of outgoing ants were evenly distributed between the two doors. We computed the index of asymmetry $I_a$, during the early and last steps of foraging—i.e. 5 and 120 min after food introduction. The asymmetry index computed after 120 min showed that experiments ranged from a symmetrical to a slightly asymmetrical use of the two entrances, with a minimal value of $I_{a120\,min} = 0.08$ and a maximal value of $I_{a120\,min} = 0.43$ (electronic supplementary material, table S1). Over the whole duration of the experiments, the asymmetry in the use of nest entrances tended to decrease, from a median $I_{a5\,min}$ index of 0.37 [0; 1] in the first 5 min to a median $I_{a120\,min}$ index of 0.1 [0.02; 0.43] at the end of the experiment (median [range], Wilcoxon signed-rank test, $p = 0.008$, $W = 44$, $n = 9$). Furthermore, we noted that the colonies of which entrances were the most unequally used at the end of the experiment were also the ones that showed the highest asymmetrical indices in the early steps of recruitment (Spearman's correlation between asymmetry indices observed after 5 and 120 min of foraging, $p = 0.036$, $\rho = 0.7$, $n = 9$). We also found that the index of asymmetry in the early outflows ($I_{a5\,min}$) was not significantly related to the initial level of asymmetry in ant densities at the two nest entrances (Spearman's correlation, $p = 0.58$, $\rho = -0.22$, $n = 9$). Furthermore, at the beginning of each trial, the ants were distributed equally at both entrance areas (binomial test, n.s.; all $p$-values > 0.05). Asymmetry in the initial spatial distribution of potential recruits in the entrance areas can thus be dismissed as an explanation for the asymmetrical use of entrances by outgoing foragers. Eventually, we evaluated the level of synchronization between outgoing flows of ants at the two entrances. The flows of outgoing ants were not strongly correlated between the two entrances as no index of synchronization $I_s$ showed a statistically significant $p$-value (permutation test with Bonferroni correction, $n = 9$, all $p > 0.0056$). This suggests that the recruitment activity which was taking place at one entrance did not influence the ants' responses at the other entrance.

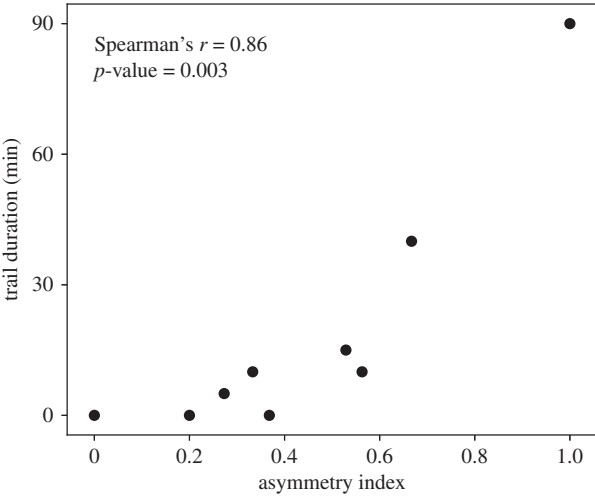

**Figure 4.** Trail duration as a function of the asymmetry of entrance use. Asymmetry was measured over the first 5 min period, before the building of any trail. Trail duration is the summation of 5 min periods where a trail is automatically detected.

## 3.2. Emergence of a foraging trail

We tested whether an additional nest entrance influenced the probability for a foraging trail to emerge as well as its duration. The automatized image analysis identified a spatial organization of foragers along a common trail in 79 out of the 432 images computed every 5 min period among the 18 experiments (for an example of trail detection see figure 3). First, we found a significant colonial effect, with some colonies being more likely to forage on a collective trail, regardless of the nest configuration. Indeed, for each colony, the total time during which a trail was detected in the one-entrance nest was significantly correlated with the one observed in the two-entrance configuration (Spearman's correlation, $p = 0.020$, $\rho = 0.75$, $n = 9$). Concerning the impact of the nest configuration on the formation of a foraging trail, eight colonies (out of nine) that were hosted in one-entrance nest showed at least one period during which a structured trail emerged, whereas only six colonies did when hosted in two-entrance nests (table 1; electronic supplementary material, figure S2). These trails emerged at times that were not significantly different for the one-entrance and the two-entrance nests (respectively, 12.5 [5; 50] min, $n = 8$, and 25 [5; 35] min, $n = 6$, after the start of the experiment, Mann–Whitney $U$-test, $p = 0.21$, $U = 17.5$; table 1, electronic supplementary material, figure S2). Interestingly, in the majority of cases (seven out of the nine tested colonies), the trail duration was equally or longer lasting in the one-entrance than in the two-entrance nest configuration. However, on average, the difference in trail duration for one-entrance nests (32.5 [5; 70] min) and two-entrance nests (12.5 [5; 90] min) were not statistically significant (table 1; Mann–Whitney $U$-test, $p = 0.24$, $U = 18$).

Finally, we found for two-entrance nests that the level of asymmetry in the use of nest entrance (as measured after 5 min of foraging) was positively correlated to the duration of the trail (Spearman's correlation, $p = 0.0017$; figure 4). The two colonies that built longer lasting trails in the two-entrance nests also showed the highest asymmetry indexes ($I_{aC8} = 1$, $I_{aC9} = 0.67$), meaning that foragers' activity was focused on one of the two nest entrances. In other words, the concentration of foraging activity on a single entrance seemed to favour the occurrence as well as the maintenance of a structured trail towards the food source. Alternatively, two-entrance nests seemed to alter the trail building process, especially when foragers equally used the two passageways.

## 3.3. Ability to reach the food source and resource exploitation

We found that the entrance configuration had a significant influence on the journeys of outgoing ants–i.e. on the percentage of individuals reaching the food source, going back to the nest or strolling in the arena (figure 5). In the case of one-entrance nests, 36% of ant individuals successfully reached the food source, 33% went back to the nest and 31% kept strolling in the arena. For the two-entrance nests, the percentage of ants reaching the food source dropped to 27%, and to 23% for those going back to the nest. By contrast, the proportion of ants that kept on strolling in the arena for more than 3 min increased to 50%. The second entrance had a significant impact on the number of ants that kept strolling in the environment (GLMM, $z = 2.712$, $p = 0.0067$) but not on the number of foragers that came back to

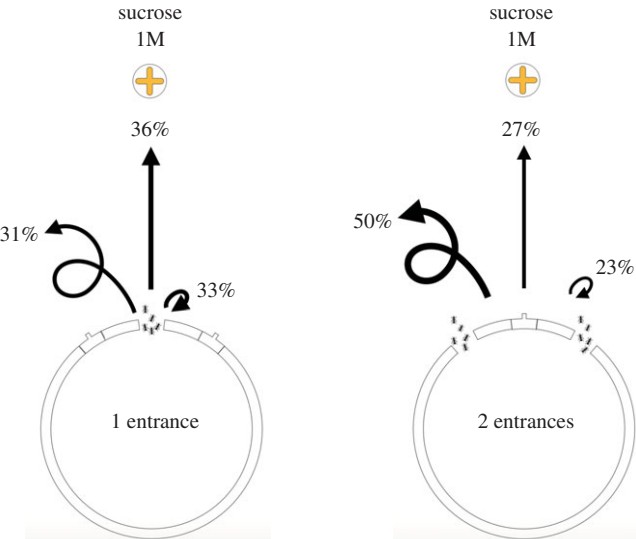

**Figure 5.** Individual follow-up of outgoing ants in one and two-entrance nests. Proportion of ants reaching the food resource, returning to the nest or that kept on strolling in the environment after 3 min of observation according to the nest entrance configuration ($n = 90$ ants per entrance configuration). The entrance configuration significantly impacts the journey of ants.

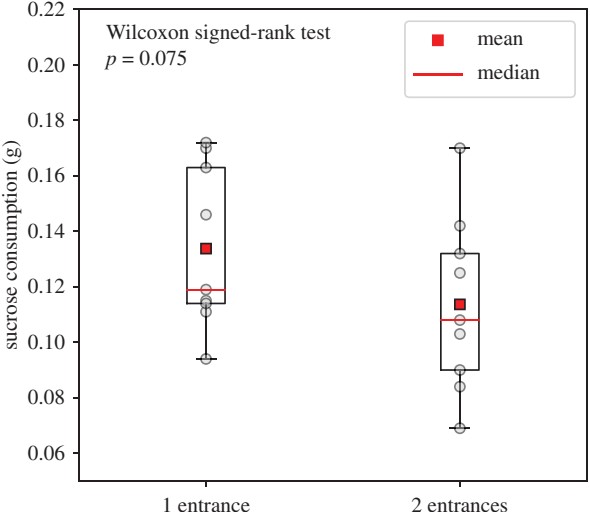

**Figure 6.** Influence of the number of nest entrances on the total amount of sucrose solution consumed at the end of the experiment. The box extends from the lower to upper quartile values of the data, with a red line at the median. The whiskers extend from the box to show the range of the data ($n = 9$, Wilcoxon signed-rank test, $p = 0.075$).

the nest (GLMM, $z = -1.483$, $p = 0.14$) or that reached the food source (GLMM, $z = -0.996$, $p = 0.32$). Finally, the journey duration was not significantly different from one nest configuration to the other among the ants that reached the food source (Mann–Whitney $U$-test, $n_1 = 32$, $n_2 = 24$, $p = 0.29$, $U = 349.5$).

We then considered the global efficiency of food exploitation at the end of the experiment. Ant colonies that were hosted in one-entrance nests consumed a median [range] of 119 [94; 172] mg of sucrose solution. When hosted in two-entrance nests, the same colonies ingested a lower median amount of 108 [69; 170] mg, although this difference in sugar consumption was not statistically significant (Wilcoxon signed-rank test, $n = 9$, $p = 0.075$; figure 6) with no influence of the treatment order (Mann–Whitney $U$-test, $n_1 = 6$, $n_2 = 3$, $p = 0.09$, $U = 16$). It is worth noticing that one colony behaved differently from the others since it consumed 1.4 times more sucrose in the two-entrance than in the one-entrance nest configuration. Interestingly, this colony was also the one that showed a highly asymmetrical use of the two entrances and that, unlike the other colonies, had a longer lasting foraging trail in the two-entrance nest configuration. When we excluded this latter colony from data analysis, we found that the ant colonies consumed a significantly

lower amount of sucrose solution when hosted in a two-entrance nest than in a one-entrance nest (Wilcoxon signed-rank test, $n = 8$, $p = 0.012$).

## 4. Discussion

Our results demonstrate that the nest structure, and particularly the interface between the nest and the environment, influences the collective exploitation of food resources by ant colonies. We found that the number of nest entrances did not influence the global mobilization of workers towards a single food source but somewhat altered the recruitment dynamics, the emergence of a foraging trail, and ultimately may affect the total amount of food harvested by the colony as a whole. Indeed, a second entrance made a common chemical trail less likely to emerge and often reduced the duration of this collective foraging structure. We suggest the following mechanism: owing to the guiding role of foraging trails, which help recruits to orient themselves towards a chosen food source, the lower occurrence of foraging structures resulted in a lower percentage of ants that successfully reached the feeder, and thus in a lower sucrose consumption for most of the two-entrance nests.

Concerning the early steps of ant mobilization, we found that the maximum outflows of recruits were slightly delayed in two-entrance nests. Given that the number of scouts that simultaneously discovered food was limited, a second nest entrance may have divided the information flow and reduced the return-rate of informed scouts at each of the two nest entrances. Because the rate of encounters with recruiters plays a key-role in the recruitment speed [15–17,19,26], this second entrance would have reduced the average level of activation that nest-mates experienced locally, thereby slowing down their mobilization outside the nest.

Our results also showed that there was no synchronicity between the flows of workers observed at the two entrances. This lack of synchronized activity suggests that the impact of recruiting scouts was restricted to the entrance that they entered and did not propagate to the other pool of workers staying near the second entrance. Thus, the two entrances should be considered as two separate units whose nest-mates behave rather independently from each other during the recruitment process.

Moreover, unlike exploring ants that quickly make use of newly open nest entrances and that easily re-distribute themselves [29], recruits seemed less able to evenly balance their outgoing flows between the different nest openings. The unbalanced use of the entrances observed during the early foraging steps, which was not owing to differences in the local density of nest-mates, could persist during the whole experiment. Once recruiters had induced an unequal mobilization of nest-mates between the two entrances, a re-equilibration of foraging activity was difficult to achieve, possibly owing to ants being 'trapped' by trails, to ants' fidelity to the entrance they first exited as well as to a limited intermingling of ants' population from the two-entrance areas.

In mass recruiting ants such as M. rubra, the chemical trail laid by recruiters not only stimulates workers to exit the nest but also guides them to the discovered food source [31]. We have shown that the emergence of a collective trail was slightly delayed, or its occurrence even altered owing to the scattering of a limited number of foragers between multiple entrances. As a result, the collective trail was replaced by a fuzzy network of interwoven individual paths. Well-drawn foraging trails may appear out of two-entrance nests only when the ants concentrated their foraging activity on one of the two entrances. This is explained by the nonlinear dependence of trail emergence on its rate of reinforcement by recruiting ants. Below a critical density of trail-laying ants, the local rate of trail reinforcement may be too small to compensate for pheromone loss by evaporation or adsorption in the substrate, thereby making a structured pattern unlikely to emerge [33]. Thus, our study confirms that the occurrence of collective foraging trails is closely linked to the existence of key locations such as the entrance area where potential recruits are clumped and undergo sufficiently high rates of interactions with recruiting ants [34,35]. Likewise, the emergence of collective foraging trails is facilitated by the concentration of ants within spatially limited zones such as areas marked with colony odour or exploratory trails [35–38].

Our study also sheds light on the high sensitivity of mass recruiting ant species to the number of nest entrances. In this recruitment process, several steps determine how ant colonies will organize and efficiently exploit food resources. First, the motivation to forage is activated by tactile and chemical stimuli emitted by recruiters within the nest boundaries, then the spatial information about the location of food sources is provided by a chemical trail laid outside the nest. These two crucial steps for a successful collective foraging are thus segregated spatially—inside versus outside the nest—and temporally—before versus after the recruits exit the nest. An efficient handover of information from the recruiters to the recruits will be best achieved when a common chemical trail can be followed

from the inside of the nest until the food target. Any environmental feature that makes this chain of activation and information transfer less efficient, such as the scattering of informed recruiters between multiple entrances, may jeopardize the emergence of collective foraging patterns.

Alternative recruitment strategies may be more resilient to an increasing number of nest entrances, such as those relying on informed leaders that physically guide recruits to the discovered food source, e.g. in tandem-running [39–43], or in group-recruitment [44–46]. Here, both the motivation to forage and the food location are directly transmitted by the leading individual to its followers. By bypassing the trail as a compulsory means to reach the food target, these leader-based recruitments are expected to be less sensitive to the number of nest openings. Likewise, colonies may be less impacted by changes in the number of entrances when the path to the resource is 'anchored' in the environment, such as the physically delineated trunk-trails in leaf-cutting ants [47] or when the foraging area is partitioned with particular trails leading to particular food patches.

From an ecological perspective, multiple nest entrances, which delay the emergence of collective trails and decrease the efficiency of social foraging, make mass recruiting ants less competitive for the exploitation of large stable resources (e.g. honeydew-producing aphids). On the other hand, multiple entrances result in having more workers strolling in the environment, thereby favouring the finding of food sources in nature. In the case of ants with an omnivorous diet, such as *M. rubra* ant species, multiple entrances should thus facilitate the discovery of small, dispersed and ephemeral resources such as insect cadavers [48]. Furthermore, multiple nest entrances may be advantageous by covering a wider foraging area, spreading out more evenly foragers across their home range and thereby maximizing food encounter rates, particularly for scattered food items [49]. Finally, in the case of polydomy, where each entrance is specifically connected to a subunit of the colony, supplementary entrances will enhance foraging efficiency, as they will favour the division of labour and thus, the efficiency of food exploitation at the colony level [50].

In insect societies, their nest structure is known to deeply impact the spatial distribution of nest-mates and therefore the way they will interact, exchange information and ultimately display collective responses [18,20]. Our study demonstrates that the nest architecture, in particular its number of entrances, can exert its influence beyond the nest boundaries, up to the interactions taking place in the environment and to the related collective foraging structures. This raises interesting questions about the functional value of physical structures for different systems of information sharing. Comparative studies would allow investigation of how the efficiency of certain foraging strategies may depend on the structure of the interface through which a society interacts with its surrounding environment.

Data accessibility. The data are accessible in the electronic supplementary material.

Authors' contribution. M.L. carried out the experiments, collected the data, analysed the data and wrote the manuscript. B.C. developed the automated detection of foraging trails and wrote the manuscript. C.D. coordinated the study, analysed the data and wrote the manuscript. All authors gave final approval for publication and agree to be held accountable for the work performed therein.

Competing interests. We have no competing interests.

Funding. This work was supported by a Belgian PhD Grant from the F.R.I.A. (Fonds pour la formation à la Recherche dans l'Industrie et dans l'Agriculture) attributed to M.L. and a Belgian Post-doctoral Grant from the Belgian National Fund for Scientific Research (F.N.R.S) attributed to B.C. C.D. is Research Director from the Belgian National Fund for Scientific Research (F.N.R.S).

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
