## [Reviewer comments · Royal Society Open Science]

Review History

RSOS-191330.R0 (Original submission)

Review form: Reviewer 1

Is the manuscript scientifically sound in its present form?

Yes

Are the interpretations and conclusions justified by the results?

No

Is the language acceptable?

Yes

Do you have any ethical concerns with this paper?

No

Have you any concerns about statistical analyses in this paper?

No

Recommendation?

Major revision is needed (please make suggestions in comments)

Comments to the Author(s)

The manuscript entitled “Multiple entrances alter foraging and information transfer in ants” is testing the role of having two vs one nest entrance in terms of foraging dynamics (e.g. trail establishment, consumption of resource). The paper is well written, and the experiments in the lab were carefully designed. However, I have concerns about the relevance of these results to the natural environment of the ants, and about the impact of these findings in our understanding of nest architecture.

First, I strongly encourage the authors to provide a better description of the nest architecture in nature, as well as the natural history of the species (L. 67 – 69). So, it is said that *M. rubra* nests usually have more than one entrance, but it is not explained whether it varies with colony size? Or under which conditions will they open another entrance? This information is very relevant because, in L. 94, it is said that the experiment was done with colonies of 300 workers, hence, would a colony of that size have more than one entrance in nature? If so, how many? Also, the authors should also clarify whether the colonies had one or multiple entrances when collected in the field.

L. 230-331. Related to my previous comment, can you explain how do you get exit rates of ~600 ants, when the colonies were supposed to be 300 workers? If the numbers refer to the number of times passing the entrances, then it should be called something like that, and not “number of workers” because it is confusing.

Also, at least in other ant species (e.g. leaf-cutters), it is unlikely that workers use different entrances when exploiting a single food source. Can the authors provide behavioral observations of the ants using more than one entrance for the same food source in the field, under natural conditions?

The implications of the natural architecture, its relation to colony size, and the likelihood of using two entrances to exploit the same resource should be discussed in depth, before affirming it is suboptimal (L. 443), especially because they almost consumed the same amount of sucrose solution. The results wouldn't be surprising if colonies of that size, usually keep just one entrance, but the workers are now divided between two exits and sources of info, but maybe that is not the case. Perhaps, a larger colony, that naturally builds two entrances, would separate the workers better between the two exits.

The result of having more workers strolling in the environment (L.322) in the two-entrances nests have important implications for finding other food sources in nature, especially because the increase did not affect the number of foragers that returned to the nest or that reached the food. Please include this in the Discussion.

Finally, I recommend shortening the discussion especially in parts where the study lacks the evidence for it (e.g. L. 426-441) and to focus it more on the relevance for nest occurring in natural environments.

Review form: Reviewer 2

Is the manuscript scientifically sound in its present form?

Yes

Are the interpretations and conclusions justified by the results?

Yes

Is the language acceptable?

Yes

Do you have any ethical concerns with this paper?

No

Have you any concerns about statistical analyses in this paper?

No

Recommendation?

Accept with minor revision (please list in comments)

Comments to the Author(s)

This paper describes a well-designed and excellently presented experiment to explore the implications of a particular aspect of nest structure (number of entrances) on the dynamics of collective foraging in an ant species.

I think the design and results are sound, and I recommend the paper for publication, after some revisions to the writing. I have several suggestions for improving the clarity and precision of the paper below – some very minor, but some a little weightier.

Line 12 – I suggest changing to “the total outflow of workers”, because from the abstract isn’t alone, it isn’t clear whether you are comparing per entrance outflow or colony-level outflow. Lines 50-57. While the information-transfer benefits of workers at the nest entrance are indeed likely to be important, this is unlikely to have been the primary adaptive benefit of the phenomenon of workers clustering in this zone. Nest defence is a major consideration for social insect species. In honeybees, for example, where information transfer occurs in the form of waggle-dancing on the dance floor, there are still ‘guard bees’ at the nest entrance. It seems most likely that ant workers congregating at nest entrances evolved this behaviour to defend the nest, and that any information benefit is secondary. I think this should at least be acknowledged in this section, which seems to imply information-transfer explains the whole story. Line 67. It would help the argument (and your reader) to provide some citations for this general statement across ant species.

Methods

I have a general issue about the methods – it is stated that the order of the treatments is randomised, but nowhere in the results is any order affect mentioned. Since the sample size is quite small (9) and most of the effect sizes very small, the difference between 4 vs 5 colonies going first in a particular treatment could actually make a difference to the results. I was surprised order wasn’t either compared and ruled out as having an effect, or included in analyses as a covariate. In particular, the order could affect colony satiation and hence motivation to forage and form a trail. In the 2-entrance treatments, some colonies formed trails and others didn’t – was this related to the order in which they experienced this treatment? Also, could the variation in how much colonies ate total be partly explained by the order of the treatments?

Line 95. ‘comprised’ is not the right word here. The nests comprise more than just the entrances. ‘included’ or ‘were provided with’ would work.

Paragraph starting line 126. Were these measurements taken from the videos, or live? And were the observers/video-scorers blind to the treatment/hypothesis?

Line 133. This is misleading – workers were not individually marked, so there is no way of knowing how many workers were mobilised. I recommend changing “total number of mobilised workers” to “total number of foraging trips”. Similarly, lines 230-231 are very misleading, talking about workers being “drained out”, suggesting no ants return during the 120 minutes! I presume this is not correct. This section should be discussing total numbers of foraging trips, not workers being drained out.

Line 164. There was only one experiment performed. Do you mean trials? Or colonies?

Line 183. I like this automated method of trail detection – well explained and a sensible approach to this difficult problem. I’m not quite sure what is meant by ‘time’ in line 183. Do you mean commencement time?

Section 2.6. Was the tracking of these ants performed blind to the hypothesis? While the nest status of the nest is obvious, observers blind the hypothesis could be used to avoid unconscious bias.

Results

Line 222. The SDs are dramatically different – 1.9 vs 24! Is this a missing decimal point, or where the 2-entrance nests really much more variable in outflow than the 1-entrance nests?

Lines 222, 224, 243, 244 – I presume you are using non-parametric tests because the data are non-normal, so presenting mean and sd is a bit odd – would be more fitting and probably informative to present median and range.

Figure 2a. Please add units to the y axis.

Lines 253-256. This is a bit confusing. Finding no difference in ant density in the entrance region when comparing right and left entrances says nothing about asymmetry. They could be highly asymmetric, but random with respect to left-right position, and overall would come out as non-significantly asymmetric if all the lefts are treated as the same category. The question is whether the level and direction of asymmetry in density is in the same level and direction as the asymmetry in outflow

Figure 5. I like this figure, but I think the variation in thickness of the arrows is confusing. The eye assumes the thicker arrows indicate more ants – but your 33% arrow is thinner than your 31% one. I suggest either making all the arrows equally thick, or making the thickness accurately proportional to the value, and stating this in the legend.

Discussion

The first paragraph oversteps the data a bit. Since the results supporting statement in line 358 were non-significant without a post-hoc removal of a particular colony, I recommend changing to “and ultimately may affect the total amount of food harvested”. Also, the sentence starting “Due to” on line 360 presents the authors’ inference as fact. This is a possible, even likely, mechanism to explain the observations, but should be presented as such, not as a fact proven in the experiment – no actual manipulation of the trails was performed, so their role can be speculated upon, but not strongly concluded. Suggest prefix sentence with “We suggest the following mechanism:”
Line 404. Sp. Should read spp.; less should read ‘lesser’

Lines 425-437. In many ant species there is spatial partitioning of the foraging area, with particular trails leading to particular areas. This might mean that in natural situations, each entrance would point to a particular zone, and thus, that the set-up here is quite unnatural, because trails from different entrances are converging into the same zone.

Lines 425-454. Another issue here is related again to defence – maybe having multiple entrances means that the colony must allocate more workers to nest-entrance defence, and that might decrease the number of foragers available. As workers weren’t marked, we don’t know if this could be happening here.

Lines 442-454. These are important caveats – and it would have been interesting to see the ‘other half’ of the experiment, in which colonies were provided with 2 food sources, to see whether in that circumstance, the 2-entrance colonies performed better than the 1-entrance colonies.

Decision letter (RSOS-191330.R0)

04-Nov-2019

Dear Dr Collignon,

The editors assigned to your paper ("Multiple nest entrances alter foraging and information transfer in ants") have now received comments from reviewers. We would like you to revise your paper in accordance with the referee and Associate Editor suggestions which can be found

below (not including confidential reports to the Editor). Please note this decision does not guarantee eventual acceptance.

Please submit a copy of your revised paper before 27-Nov-2019. Please note that the revision deadline will expire at 00.00am on this date. If we do not hear from you within this time then it will be assumed that the paper has been withdrawn. In exceptional circumstances, extensions may be possible if agreed with the Editorial Office in advance. We do not allow multiple rounds of revision so we urge you to make every effort to fully address all of the comments at this stage. If deemed necessary by the Editors, your manuscript will be sent back to one or more of the original reviewers for assessment. If the original reviewers are not available, we may invite new reviewers.

- Data accessibility

<http://datadryad.org/submit?journalID=RSOS&manu=RSOS-191330>

- Competing interests

- Authors' contributions

All submissions, other than those with a single author, must include an Authors' Contributions section which individually lists the specific contribution of each author. The list of Authors should meet all of the following criteria; 1) substantial contributions to conception and design, or

acquisition of data, or analysis and interpretation of data; 2) drafting the article or revising it critically for important intellectual content; and 3) final approval of the version to be published.

- Acknowledgements

- Funding statement

Kind regards,
Andrew Dunn
Senior Publishing Editor
Royal Society Open Science
openscience@royalsociety.org

on behalf of Professor Wen-Xu Wang (Associate Editor) and Kevin Padian (Subject Editor)
openscience@royalsociety.org

Comments to Author:

Reviewers' Comments to Author:

Reviewer: 1

Comments to the Author(s)

The manuscript entitled "Multiple entrances alter foraging and information transfer in ants" is testing the role of having two vs one nest entrance in terms of foraging dynamics (e.g. trail establishment, consumption of resource). The paper is well written, and the experiments in the lab were carefully designed. However, I have concerns about the relevance of these results to the natural environment of the ants, and about the impact of these findings in our understanding of nest architecture.

First, I strongly encourage the authors to provide a better description of the nest architecture in nature, as well as the natural history of the species (L. 67 – 69). So, it is said that *M. rubra* nests usually have more than one entrance, but it is not explained whether it varies with colony size? Or under which conditions will they open another entrance? This information is very relevant because, in L. 94, it is said that the experiment was done with colonies of 300 workers, hence, would a colony of that size have more than one entrance in nature? If so, how many? Also, the authors should also clarify whether the colonies had one or multiple entrances when collected in the field.

L. 230-331. Related to my previous comment, can you explain how do you get exit rates of ~600 ants, when the colonies were supposed to be 300 workers? If the numbers refer to the number of

times passing the entrances, then it should be called something like that, and not “number of workers” because it is confusing.

Also, at least in other ant species (e.g. leaf-cutters), it is unlikely that workers use different entrances when exploiting a single food source. Can the authors provide behavioral observations of the ants using more than one entrance for the same food source in the field, under natural conditions?

The implications of the natural architecture, its relation to colony size, and the likelihood of using two entrances to exploit the same resource should be discussed in depth, before affirming it is suboptimal (L. 443), especially because they almost consumed the same amount of sucrose solution. The results wouldn't be surprising if colonies of that size, usually keep just one entrance, but the workers are now divided between two exits and sources of info, but maybe that is not the case. Perhaps, a larger colony, that naturally builds two entrances, would separate the workers better between the two exits.

The result of having more workers strolling in the environment (L.322) in the two-entrances nests have important implications for finding other food sources in nature, especially because the increase did not affect the number of foragers that returned to the nest or that reached the food. Please include this in the Discussion.

Finally, I recommend shortening the discussion especially in parts where the study lacks the evidence for it (e.g. L. 426-441) and to focus it more on the relevance for nest occurring in natural environments.

Reviewer: 2

Comments to the Author(s)

This paper describes a well-designed and excellently presented experiment to explore the implications of a particular aspect of nest structure (number of entrances) on the dynamics of collective foraging in an ant species.

I think the design and results are sound, and I recommend the paper for publication, after some revisions to the writing. I have several suggestions for improving the clarity and precision of the paper below – some very minor, but some a little weightier.

Line 12 – I suggest changing to “the total outflow of workers”, because from the abstract isn't alone, it isn't clear whether you are comparing per entrance outflow or colony-level outflow. Lines 50-57. While the information-transfer benefits of workers at the nest entrance are indeed likely to be important, this is unlikely to have been the primary adaptive benefit of the phenomenon of workers clustering in this zone. Nest defence is a major consideration for social insect species. In honeybees, for example, where information transfer occurs in the form of waggle-dancing on the dance floor, there are still ‘guard bees’ at the nest entrance. It seems most likely that ant workers congregating at nest entrances evolved this behaviour to defend the nest, and that any information benefit is secondary. I think this should at least be acknowledged in this section, which seems to imply information-transfer explains the whole story.

Line 67. It would help the argument (and your reader) to provide some citations for this general statement across ant species.

Methods

I have a general issue about the methods – it is stated that the order of the treatments is randomised, but nowhere in the results is any order affect mentioned. Since the sample size is quite small (9) and most of the effect sizes very small, the difference between 4 vs 5 colonies going first in a particular treatment could actually make a difference to the results. I was surprised order wasn't either compared and ruled out as having an effect, or included in analyses

as a covariate. In particular, the order could affect colony satiation and hence motivation to forage and form a trail. In the 2-entrance treatments, some colonies formed trails and others didn't – was this related to the order in which they experienced this treatment? Also, could the variation in how much colonies ate total be partly explained by the order of the treatments? Line 95. 'comprised' is not the right word here. The nests comprise more than just the entrances. 'included' or 'were provided with' would work.

Paragraph starting line 126. Were these measurements taken from the videos, or live? And were the observers/video-scorers blind to the treatment/hypothesis?

Line 133. This is misleading – workers were not individually marked, so there is no way of knowing how many workers were mobilised. I recommend changing "total number of mobilised workers" to "total number of foraging trips". Similarly, lines 230-231 are very misleading, talking about workers being "drained out", suggesting no ants return during the 120 minutes! I presume this is not correct. This section should be discussing total numbers of foraging trips, not workers being drained out.

Line 164. There was only one experiment performed. Do you mean trials? Or colonies?

Line 183. I like this automated method of trail detection – well explained and a sensible approach to this difficult problem. I'm not quite sure what is meant by 'time' in line 183. Do you mean commencement time?

Section 2.6. Was the tracking of these ants performed blind to the hypothesis? While the nest status of the nest is obvious, observers blind the hypothesis could be used to avoid unconscious bias.

Results

Line 222. The SDs are dramatically different – 1.9 vs 24! Is this a missing decimal point, or where the 2-entrance nests really much more variable in outflow than the 1-entrance nests?

Lines 222, 224, 243, 244 – I presume you are using non-parametric tests because the data are non-normal, so presenting mean and sd is a bit odd – would be more fitting and probably informative to present median and range.

Figure 2a. Please add units to the y axis.

Lines 253-256. This is a bit confusing. Finding no difference in ant density in the entrance region when comparing right and left entrances says nothing about asymmetry. They could be highly asymmetric, but random with respect to left-right position, and overall would come out as non-significantly asymmetric if all the lefts are treated as the same category. The question is whether the level and direction of asymmetry in density is in the same level and direction as the asymmetry in outflow

Figure 5. I like this figure, but I think the variation in thickness of the arrows is confusing. The eye assumes the thicker arrows indicate more ants – but your 33% arrow is thinner than your 31% one. I suggest either making all the arrows equally thick, or making the thickness accurately proportional to the value, and stating this in the legend.

Discussion

The first paragraph oversteps the data a bit. Since the results supporting statement in line 358 were non-significant without a post-hoc removal of a particular colony, I recommend changing to "and ultimately may affect the total amount of food harvested". Also, the sentence starting "Due to" on line 360 presents the authors' inference as fact. This is a possible, even likely, mechanism to explain the observations, but should be presented as such, not as a fact proven in the experiment – no actual manipulation of the trails was performed, so their role can be speculated upon, but not strongly concluded. Suggest prefix sentence with "We suggest the following mechanism:"

Line 404. Sp. Should read spp.; less should read 'lesser'

Lines 425-437. In many ant species there is spatial partitioning of the foraging area, with particular trails leading to particular areas. This might mean that in natural situations, each entrance would point to a particular zone, and thus, that the set-up here is quite unnatural, because trails from different entrances are converging into the same zone.

Lines 425-454. Another issue here is related again to defence – maybe having multiple entrances means that the colony must allocate more workers to nest-entrance defence, and that might decrease the number of foragers available. As workers weren't marked, we don't know if this could be happening here.

Lines 442-454. These are important caveats – and it would have been interesting to see the ‘other half’ of the experiment, in which colonies were provided with 2 food sources, to see whether in that circumstance, the 2-entrance colonies performed better than the 1-entrance colonies.

Author's Response to Decision Letter for (RSOS-191330.R0)

See Appendix A.

RSOS-191330.R1 (Revision)

Review form: Reviewer 2

Is the manuscript scientifically sound in its present form?

Yes

Are the interpretations and conclusions justified by the results?

Yes

Is the language acceptable?

Yes

Do you have any ethical concerns with this paper?

No

Have you any concerns about statistical analyses in this paper?

No

Recommendation?

Accept as is

Comments to the Author(s)

I am satisfied that the authors have responded to and dealt with all my concerns and those of the other reviewer.

Review form: Reviewer 3

Is the manuscript scientifically sound in its present form?

Yes

Are the interpretations and conclusions justified by the results?

Yes

Is the language acceptable?

Yes

Do you have any ethical concerns with this paper?

No

Have you any concerns about statistical analyses in this paper?

Yes

Recommendation?

Accept with minor revision (please list in comments)

Comments to the Author(s)

This work tests the idea that the number of nest entrances regulates social exchanges between ant foragers and inner-nest workers, and hence influences the foraging efficiency of the whole colony. To do this, the authors compared the foraging responses of *Myrmica rubra* ant colonies settled in one-entrance versus two-entrance nests. They found that the total outflows of workers exploiting a sucrose food source were similar regardless of the number of nest entrances. However, in the two-entrance nests, the launching of recruitment was delayed, a pheromone trail was less likely to emerge between the nest and the food source, and recruits were less likely to reach the food source. As a result, an additional nest entrance through which information could transit decreased the efficiency of social foraging and ultimately led to a lower amount of retrieved food.

This is a second version of the manuscript, which includes detailed responses to previous reviewer comments. Since this is my first revisions of the paper, I will focus only in the paper itself rather than on the author's responses.

I consider that the paper is well written, and the lab experiments were nice- designed and very good presented in the text. The results and their interpretation are sound. I do not have major comments, only minor suggestions that I think may improve the clarity of the manuscript.

Minor comments

L. 60. The rate at which recruits leave the nest also depends on the design of the nest entrances (see Rodríguez-Planes & Alejandro G. Farji-Brener, 2019). This reference may be of interest in the discussion developed at the lines 463-467.

L. 130. So, the same colony was exposed to the two experimental procedures? Please emphasize that this is the experimental design to avoid confusion among the readers.

L. 228. Worker mobilization. In the comparison between one versus two nest-entrances situation it is unclear for me if the value from the two nest entrances situation is the mean of the two entrances or the sum of the two entrances. Please clarify and justify

L.305. Why here a Mann -Whitney test is used instead the Wilcoxon- paired tests? If you are comparing the trail duration between the two configurations (one and two entrances) of the same colony, paired t test or its non-parametric equivalent (Wilcoxon) is more appropriated.

References

Rodríguez-Planes & Alejandro G. Farji-Brener. 2019. Extended phenotypes and foraging restrictions: ant nest entrances and resource ingress in leaf-cutting ants. *Biotropica* 51:178-185. <https://doi.org/10.1111/btp.12630>.

Decision letter (RSOS-191330.R1)

02-Jan-2020

Dear Dr Collignon:

On behalf of the Editors, I am pleased to inform you that your Manuscript RSOS-191330.R1 entitled "Multiple nest entrances alter foraging and information transfer in ants" has been accepted for publication in Royal Society Open Science subject to minor revision in accordance with the referee suggestions. Please find the referees' comments at the end of this email.

The reviewers and Subject Editor have recommended publication, but also suggest some minor revisions to your manuscript. Therefore, I invite you to respond to the comments and revise your manuscript.

- Ethics statement

- Data accessibility

<http://datadryad.org/submit?journalID=RSOS&manu=RSOS-191330.R1>

- Competing interests

- Authors' contributions

AB carried out the molecular lab work, participated in data analysis, carried out sequence alignments, participated in the design of the study and drafted the manuscript; CD carried out the statistical analyses; EF collected field data; GH conceived of the study, designed the study,

coordinated the study and helped draft the manuscript. All authors gave final approval for publication.

- Acknowledgements

- Funding statement

Because the schedule for publication is very tight, it is a condition of publication that you submit the revised version of your manuscript before 11-Jan-2020. Please note that the revision deadline will expire at 00.00am on this date. If you do not think you will be able to meet this date please let me know immediately.

on behalf of Professor Wen-Xu Wang (Associate Editor) and Kevin Padian (Subject Editor)
 openscience@royalsociety.org

Reviewer comments to Author:
 Reviewer: 2

Comments to the Author(s)
 I am satisfied that the authors have responded to and dealt with all my concerns and those of the other reviewer.

Reviewer: 3

Comments to the Author(s)
 This work tests the idea that the number of nest entrances regulates social exchanges between ant foragers and inner-nest workers, and hence influences the foraging efficiency of the whole colony. To do this, the authors compared the foraging responses of *Myrmica rubra* ant colonies settled in one-entrance versus two-entrance nests. They found that the total outflows of workers exploiting a sucrose food source were similar regardless of the number of nest entrances. However, in the two-entrance nests, the launching of recruitment was delayed, a pheromone trail was less likely to emerge between the nest and the food source, and recruits were less likely to reach the food source. As a result, an additional nest entrance through which information could transit decreased the efficiency of social foraging and ultimately led to a lower amount of retrieved food.

This is a second version of the manuscript, which includes detailed responses to previous reviewer comments. Since this is my first revisions of the paper, I will focus only in the paper itself rather than on the author's responses.

I consider that the paper is well written, and the lab experiments were nice- designed and very good presented in the text. The results and their interpretation are sound. I do not have major comments, only minor suggestions that I think may improve the clarity of the manuscript.

Minor comments

L. 60. The rate at which recruits leave the nest also depends on the design of the nest entrances (see Rodríguez-Planes & Alejandro G. Farji-Brener, 2019). This reference may be of interest in the discussion developed at the lines 463-467.

L. 130. So, the same colony was exposed to the two experimental procedures? Please emphasize that this is the experimental design to avoid confusion among the readers.

L. 228. Worker mobilization. In the comparison between one versus two nest-entrances situation

it is unclear for me if the value from the two nest entrances situation is the mean of the two entrances or the sum of the two entrances. Please clarify and justify

L.305. Why here a Mann -Whitney test is used instead the Wilcoxon- paired tests? If you are comparing the trail duration between the two configurations (one and two entrances) of the same colony, paired t test or its non-parametric equivalent (Wilcoxon) is more appropriated.

References

Rodríguez-Planes & Alejandro G. Farji-Brener. 2019. Extended phenotypes and foraging restrictions: ant nest entrances and resource ingress in leaf-cutting ants. *Biotropica* 51:178–185. <https://doi.org/10.1111/btp.12630>.

Author's Response to Decision Letter for (RSOS-191330.R1)

See Appendix B.

Decision letter (RSOS-191330.R2)

17-Jan-2020

Dear Dr Collignon,

It is a pleasure to accept your manuscript entitled "Multiple nest entrances alter foraging and information transfer in ants" in its current form for publication in Royal Society Open Science. The comments of the reviewer(s) who reviewed your manuscript are included at the foot of this letter.

on behalf of Professor Wen-Xu Wang (Associate Editor) and Kevin Padian (Subject Editor)
openscience@royalsociety.org

Appendix A

Referee #1

The manuscript entitled “Multiple entrances alter foraging and information transfer in ants” is testing the role of having two vs one nest entrance in terms of foraging dynamics (e.g. trail establishment, consumption of resource). The paper is well written, and the experiments in the lab were carefully designed. However, I have concerns about the relevance of these results to the natural environment of the ants, and about the impact of these findings in our understanding of nest architecture.

*First, I strongly encourage the authors to provide a better description of the nest architecture in nature, as well as the natural history of the species (L. 67 – 69). So, it is said that *M. rubra* nests usually have more than one entrance, but it is not explained whether it varies with colony size? Or under which conditions will they open another entrance? This information is very relevant because, in L. 94, it is said that the experiment was done with colonies of 300 workers, hence, would a colony of that size have more than one entrance in nature? If so, how many? Also, the authors should also clarify whether the colonies had one or multiple entrances when collected in the field.*

We agree with referee #1 that these points actually deserve to be better explained. We carried out a field study where we observed changes in the number of openings of *M. rubra* nests for several weeks. We also tracked to which extent one or several entrances were linked to food sources that were introduced in the environment. These field data are presented and discussed in another submitted paper. However, to follow referee 1 advice, we have added a few sentences that are describing the natural nests of *M. rubra* (lines 70-72 and lines 90-94).

Unfortunately, we have no data about the correlation between the colony size and the number of nest entrances. It should however be mentioned that the whole ant population is distributed among several nest subunits and that the foraging response (e.g. the number of foraging trips) depends mainly on the population of ants housed in the superficial chambers of the nest. Therefore, if the study aims to test the impact of nest entrance on foraging, it seems reasonable to use colonies containing a few hundredths of workers since it is in the same order of magnitude as the population size observed when collecting ant colonies in the field.

L. 230-331. Related to my previous comment, can you explain how do you get exit rates of ~600 ants, when the colonies were supposed to be 300 workers? If the numbers refer to the number of times passing the entrances, then it should be called something like that, and not “number of workers” because it is confusing.

Thanks for this comment. We are now talking about “total number of foraging trips”

Also, at least in other ant species (e.g. leaf-cutters), it is unlikely that workers use different entrances when exploiting a single food source. Can the authors provide behavioural observations of the ants using more than one entrance for the same food source in the field, under natural conditions?

We agree with referee #1 that large polydomous nests or nests with multiple entrances that are separated by several tenths of centimetres are unlikely to use different entrances when exploiting a single food source. However, in the case of *M. rubra* nests, one can see entrances that are separated by only a few cm (as in the experimental nest used in the present study). In this latter case, we observed that different nest entrances may be used by ants that reached the same food source. This information was added in the MS (lines 74-76).

The implications of the natural architecture, its relation to colony size, and the likelihood of using two entrances to exploit the same resource should be discussed in depth, before affirming it is suboptimal (L. 443), especially because they almost consumed the same amount of sucrose solution. The results wouldn't be surprising if colonies of that size, usually keep just one entrance, but the workers are now divided between two exits and sources of info, but maybe that is not the case. Perhaps, a larger colony, that naturally builds two entrances, would separate the workers better between the two exits.

In our field study of natural nests of *M. rubra*, it was quite usual to observe ants exiting from several active entrances spaced by a few centimetres (personal observations). However, as above-mentioned, we do not precisely know how many individuals are associated to each nest exit and we cannot reliably relate the natural nest architecture to the colony size. As regards the functional value of the nest architecture, we agree with referee #1 that speaking about optimality is too hasty and not appropriate. This would deserve further studies about nest architecture, population size in relation with a characterization of available resources. We re-edited the § (lines 444-451) accordingly (namely by removing the “suboptimality” idea).

The result of having more workers strolling in the environment (L.322) in the two-entrances nests have important implications for finding other food sources in nature, especially because the increase did not affect the number of foragers that returned to the nest or that reached the food. Please include this in the Discussion.

Thanks for the suggestion that we added in the revised paragraph (line 447-451)

Finally, I recommend shortening the discussion especially in parts where the study lacks the evidence for it (e.g. L. 426-441) and to focus it more on the relevance for nest occurring in natural environments.

Following referee #1 suggestion, we shortened the discussion by approximately 15 lines. In particular, we removed the last sentences of the paragraph dealing about the impact of entrances' number on alternative types of recruitment.

Reviewer: #2

This paper describes a well-designed and excellently presented experiment to explore the implications of a particular aspect of nest structure (number of entrances) on the dynamics of collective foraging in an ant species. I think the design and results are sound, and I recommend the paper for publication, after some revisions to the writing. I have several suggestions for improving the clarity and precision of the paper below – some very minor, but some a little weightier.

We thank the referee for this positive comment.

Line 12 – I suggest changing to “the total outflow of workers”, because from the abstract isn't alone, it isn't clear whether you are comparing per entrance outflow or colony-level outflow.

Thanks, we made these changes wherever needed.

Lines 50-57. While the information-transfer benefits of workers at the nest entrance are indeed likely to be important, this is unlikely to have been the primary adaptive benefit of the phenomenon of workers clustering in this zone. Nest defence is a major consideration for social insect species. In honeybees, for example, where information transfer occurs in the form of waggle-dancing on the dance floor, there are still ‘guard bees’ at the nest entrance. It seems most likely that ant workers congregating at nest entrances evolved this behaviour to defend the nest, and that any information benefit is secondary. I think this should at least be acknowledged in this section, which seems to imply information-transfer explains the whole story.

We agree with referee 2 that workers at the nest entrance also play a key role in nest defence as well as in sanitary control of incoming nestmates. In the revised manuscript, we better acknowledge about the functional value of workers' aggregates at nest entrances in terms of nest defence against predators, competitors or even pathogens. Lines 49-56 were reedited and new references were added.

Line 67. It would help the argument (and your reader) to provide some citations for this general statement across ant species.

Our intention was not to state that *all* ant species inhabit nests with multiple entrances, but rather that it is common to find multiple exits in ant nest. Therefore, we are now more cautious in the revised manuscript and we added additional information about the number of entrances that can be found in *M. rubra* nests at lines 69-72.

Methods

I have a general issue about the methods – it is stated that the order of the treatments is randomised, but nowhere in the results is any order affect mentioned. Since the sample size is quite small (9) and most of the effect sizes very small, the difference between 4 vs 5 colonies going first in a particular treatment could actually make a difference to the results. I was surprised order wasn't either compared and ruled out as having an effect, or included in analyses as a covariate. In particular, the order could affect colony satiation and hence motivation to forage and form a trail. In the 2-entrance treatments, some colonies formed trails and others didn't – was this related to the order in which they experienced this treatment? Also, could the variation in how much colonies ate total be partly explained by the order of the treatments?

To make sure that the level of colony satiation was similar for each treatment (nests with one or two entrances), ant colonies were always subjected to the same treatment before a trial. The ants had access to a sucrose solution up to two days before the experimental day. Then, they were starved for 48h and had only access to a water tube until the start of the experiment. At the end of the trial, the ants had again access to a sucrose solution for at least four days before being starved and undergoing the second trials. This experimental procedure ensured that the ants were equally starved before each trial.

Following referee comment, we tested whether the order of treatment had an impact on food consumption. To do so, we compared the difference of food harvested between the first and second trial for colonies that first received the “one-entrance” treatment against the same measure for colonies that first received the “two-entrance” treatment. We found out that the order

of the treatments had no impact on the difference of food harvested (Mann-Whitney, $U = 16$, $p = 0.09$). This information has been added in the revised MS.

Line 95. 'comprised' is not the right word here. The nests comprise more than just the entrances. 'included' or 'were provided with' would work.

Thanks for the suggestion. We made the change.

Paragraph starting line 126. Were these measurements taken from the videos, or live? And were the observers/video-scorers blind to the treatment/hypothesis?

These measures were taken from the videos. They were done by the first author who could see the nest configuration and thus was not blind to treatment. However, as these measures were counts, they can be considered as unambiguous. We specified in the MS that measures were done on videos at line135.

Line 133. This is misleading – workers were not individually marked, so there is no way of knowing how many workers were mobilised. I recommend changing “total number of mobilized workers” to “total number of foraging trips”. Similarly, lines 230-231 are very misleading, talking about workers being “drained out”, suggesting no ants return during the 120 minutes! I presume this is not correct. This section should be discussing total numbers of foraging trips, not workers being drained out.

Thanks to referee 2 comment, we now realize that the sentence was misleading. We made the suggested corrections.

Line 164. There was only one experiment performed. Do you mean trials? Or colonies?

Actually we meant trials. We changed the sentence (line 171) to clarify that these p-values were computed for each replicate of the experiment.

Line 183. I like this automated method of trail detection – well explained and a sensible approach to this difficult problem. I'm not quite sure what is meant by 'time' in line 183. Do you mean commencement time?

Thanks for this comment. Actually, we meant the time at which a trail first emerged. We revised the sentence (line 192-193) in order to make it clear.

Section 2.6. Was the tracking of these ants performed blind to the hypothesis? While the nest status of the nest is obvious, observers blind the hypothesis could be used to avoid unconscious bias.

The tracking of ants was performed on video recordings by the first author who was not blind to the hypothesis. However, the criteria used to define the ant destination were not ambiguous, thus left little opportunity for a biased interpretation.

Results

Line 222. The SDs are dramatically different – 1.9 vs 24! Is this a missing decimal point, or where the 2-entrance nests really much more variable in outflow than the 1-entrance nests?

Thanks for noticing. Actually, the SD values were similar for the two configuration of nest entrance with a high intercolonial variability in the baseline outflows of workers (before food introduction). In the revised MS, the mean and SD values have been replaced by the median and range values as suggested in the comment below.

Lines 222, 224, 243, 244 – I presume you are using non-parametric tests because the data are non-normal, so presenting mean and sd is a bit odd – would be more fitting and probably informative to present median and range.

Based on referee suggestion, we now provide the median and range values at these lines and in table 1. To remain coherent, we also provide median and range (instead of mean) for the asymmetry index, time of trail emergence and trail duration.

Figure 2a. Please add units to the y axis.

We added the units on the Y axis (here number of ants/5min) of Figure 2a and we also changed the caption of Y axis of figure 2B (here total number of foraging trips).

Lines 253-256. This is a bit confusing. Finding no difference in ant density in the entrance region when comparing right and left entrances says nothing about asymmetry. They could be highly asymmetric, but random with respect to left-right position, and overall would come out as non-significantly asymmetric if all the lefts are treated as the same category. The question is whether the level and direction of asymmetry in density is in the same level and direction as the asymmetry in outflow.

Referee 2 is right. To meet this insightful remark, we tested whether the level of asymmetry in the outflows during the first 5 minutes was related to the same level of asymmetry in ant density at nest entrances. We thus calculated the Spearman correlation value, which was not significant ($r=-0.22$, $p=0.58$). This confirms that the observed flows' asymmetry did not outcome from an asymmetry in the initial conditions of ants' distribution at the entrances. Furthermore, for each trial, we tested whether the ants distributed themselves equally at the two nest entrances by carrying out a binomial test that turned out to be non-significant for each trial. These statistical analyses were added at lines 274-280 in the revised MS.

Figure 5. I like this figure, but I think the variation in thickness of the arrows is confusing. The eye assumes the thicker arrows indicate more ants – but your 33% arrow is thinner than your 31% one. I suggest either making all the arrows equally thick, or making the thickness accurately proportional to the value, and stating this in the legend.

Thank you for your remark. The thickness of the arrows is now proportional to the percentage of foragers.

Discussion

The first paragraph oversteps the data a bit. Since the results supporting statement in line 358 were non-significant without a post-hoc removal of a particular colony, I recommend changing to “and ultimately may affect the total amount of food harvested”. Also, the sentence starting “Due to” on line 360 presents the authors' inference as fact. This is a possible, even likely, mechanism to explain the observations, but should be presented as such, not as a fact proven in the experiment – no actual manipulation of the trails was performed, so their role can be speculated upon, but not strongly concluded. Suggest prefix sentence with “We suggest the following mechanism:”

As suggested by referee 2, we are now more cautious in our statements and inference about the benefit in terms of harvested food. We added the changes suggested by referee 2 (lines 371, 374). Many thanks for these suggestions.

Line 404. Sp. Should read spp.; less should read 'lesser'

These lines have been deleted in the shortened discussion (as required by referee 1)

Lines 425-437. In many ant species there is spatial partitioning of the foraging area, with particular trails leading to particular areas. This might mean that in natural situations, each entrance would point to a particular zone, and thus, that the set-up here is quite unnatural, because trails from different entrances are converging into the same zone.

Actually, in ant species such as leaf-cutting or harvester ants, multiple nest entrances are related to a partitioning of the foraging area, with trails leading to separated food patches. In the present case, we would like to draw the attention on the smaller distance separating the multiple entrances of *M. rubra* colonies. Personal field observations showed that nest entrance may be separated only by a few centimetres and that foragers exiting from different entrances can be observed exploiting the same food source. To meet referee 2s comment, we added lines 74-76 to describe our field observation on *M. rubra* nests and we revised line 442 emphasizing on the case of ant species that show a spatial partitioning of their foraging area

Lines 425-454. Another issue here is related again to defence – maybe having multiple entrances means that the colony must allocate more workers to nest-entrance defence, and that might decrease the number of foragers available. As workers weren't marked, we don't know if this could be happening here.

Having multiple entrance is actually expected to result in a higher number of workers involved in guarding and defence. However, in the present case, we observed the same density of ants in the two nest configurations and the total number of foraging trips were similar between one-entrance and two-entrance nests. These two observations strongly suggest that the colony investment in foraging did not decrease with the addition of a second entrance.

Lines 442-454. These are important caveats – and it would have been interesting to see the ‘other half’ of the experiment, in which colonies were provided with 2 food sources, to see whether in that circumstance, the 2-entrance colonies performed better than the 1-entrance colonies.

We fully agree with referee 2 that the impact of multiple entrances on the exploitation of multiple food sources deserve to be investigated. This work is in progress but we first wanted to investigate into details a less complex situation with a single food target.

Appendix B

Referee #3

Minor comments:

L. 60. The rate at which recruits leave the nest also depends on the design of the nest entrances (see Rodríguez-Planes & Alejandro G. Farji-Brener, 2019). This reference may be of interest in the discussion developed at the lines 463-467.

Thank you, this reference has been added in our manuscript (line 63).

L. 130. So, the same colony was exposed to the two experimental procedures? Please emphasize that this is the experimental design to avoid confusion among the readers.

We clarified the experimental procedure and specified that each colony underwent the two treatments (lines 122 and 136).

L. 228. Worker mobilization. In the comparison between one versus two nest-entrances situation it is unclear for me if the value from the two nest entrances situation is the mean of the two entrances or the sum of the two entrances. Please clarify and justify.

When comparing the two nest configurations (one or two entrances, e.g. Fig 2), we computed the total outflow of ants (i.e. the sum of the two entrances in the case of two-entrance nests). To clarify, we added the missing "total" when needed (lines 237, 242, 247).

L.305. Why here a Mann –Whitney test is used instead the Wilcoxon- paired tests? If you are comparing the trail duration between the two configurations (one and two entrances) of the same colony, paired t test or its non-parametric equivalent (Wilcoxon) is more appropriated.

We agree with the reviewer that since the same colonies experienced the two treatments, paired tests are more appropriate (as we used them along our manuscript). Unfortunately for the measures of the trail duration and time of appearance, we could not use these tests. Indeed, for some colonies, a trail was only observed in one of the two treatments. In these cases, the values measured in the one-entrance condition (for example) could not be paired with a value in the two-entrance condition (a trail that does not appear has no time of appearance and no duration). Therefore, we had to perform Mann-Whitney U tests for these two measures.